# Development of new materials for electrothermal metals using data driven and machine learning

**Chengqun Zhou**[1☯], **Muyang Pei**[2☯], **Chao Wu**[1], **Degang Xu**[3,4], **Qiang Peng**[2]*, **Guoai He**[2,3]*

**1** Luoyang Institute of Science and Technology, School of Electrical Engineering and Automation, Luoyang, China, **2** Light alloy Research Institute, Central South University, Changsha, China, **3** National Key Laboratory for Precision Manufacturing of Extreme Service Performance, Central South University, Changsha, China, **4** Automation Institute, Central South University, Changsha, China

☯ These authors contributed equally to this work.
* heguoai@csu.edu.cn (GH); 213812031@csu.edu.cn (QP)

**Data Availability Statement:** All relevant data are within the paper and its Supporting Information files.

**Funding:** This work was Funded by Shenzhen Zhuolineng Technology Co., Ltd (HKF202200086);

## Abstract

After adopting a combined approach of data-driven methods and machine learning, the prediction of material performance and the optimization of composition design can significantly reduce the development time of materials at a lower cost. In this research, we employed four machine learning algorithms, including linear regression, ridge regression, support vector regression, and backpropagation neural networks, to develop predictive models for the electrical performance data of titanium alloys. Our focus was on two key objectives: resistivity and the temperature coefficient of resistance (TCR). Subsequently, leveraging the results of feature selection, we conducted an analysis to discern the impact of alloying elements on these two electrical properties. The prediction results indicate that for the resistivity data prediction task, the radial basis function kernel-based support vector machine model performs the best, with a correlation coefficient above 0.995 and a percentage error within 2%, demonstrating high predictive capability. For the TCR data prediction task, the best-performing model is a backpropagation neural network with two hidden layers, also with a correlation coefficient above 0.995 and a percentage error within 3%, demonstrating good generalization ability. The feature selection results using random forest and Xgboost indicate that Al and Zr have a significant positive effect on resistivity, while Al, Zr, and V have a significant negative effect on TCR. The conclusion of the composition optimization design suggests that to achieve both high resistivity and TCR, it is recommended to set the Al content in the range of 1.5% to 2% and the Zr content in the range of 2.5% to 3%.

## Introduction

In the realm of new material development, the challenge of achieving specific target performance has spurred a revolutionary transformation in material research, rendering traditional trial-and-error methods insufficient. These conventional approaches rely on limited experimental data to discretely adjust material compositions and process parameters in the quest for

National Natural Science Foundation of Hunan province (2022JJ30721).

**Competing interests:** The authors have declared that no competing interests exist.

optimal material performance. Nevertheless, it is evident that this research and development methodology is time-consuming and costly. In an effort to expedite material development and introduce more efficient research approaches, the United States initiated the Materials Genome Initiative (MGI) in 2011[1], with the aim of significantly reducing the material development cycle.

In recent years, data-driven and machine learning methods have gradually emerged as prominent players in addressing real-world engineering challenges [2–10]. The application of artificial intelligence, machine learning, and deep learning technologies in the field of materials science has garnered significant attention, as these methods are widely adopted for material discovery and design [11]. Logan et al. [12] emphasized that the process of using machine learning methods in materials informatics comprises three key elements: material data, material descriptors, and machine learning algorithms suitable for prediction. Davoodi et al. [13] employed multiple machine learning models to accurately predict the hydrogen (H2) absorption percentage in porous carbon media (PCM), offering critical insights for efficient H2 storage. Bruno et al. [14] utilized a machine learning kernel regression model to predict the electronic properties and elastic performance of materials, along with proposing an Exhaustive Enumeration algorithm for material reverse design. The majority of the methods mentioned above have effectively utilized suitable machine learning algorithms, coupled with selective feature engineering, to achieve satisfactory prediction and design outcomes.

The objective of this study is to develop electrically conductive metallic materials with high resistivity and temperature coefficient of resistance (TCR). In the domain of alloy electrical performance, while research on resistivity is extensive, investigations into the temperature coefficient of resistance (TCR) of alloys remain relatively limited. However, both resistivity and TCR are critical indicators of alloy electrical performance. Faced with the vast parameter space of multicomponent alloys, traditional experimental methods are practically incapable of encompassing all conceivable alloy combinations. Consequently, this study introduces data-driven and machine learning approaches to address the challenges of compositional optimization design, thereby accelerating the development of new materials.

The remaining sections of the introduction are as follows: Section 2 will introduce various machine learning algorithm models, encompassing predictive modeling and feature selection. Section 3 will compare a series of experimental results to determine the optimal predictive model and discuss and analyze compositional optimization. Lastly, Section 4 will summarize the work and propose future research directions.

## Method

This paper presents a three-step data-driven and machine learning-based workflow for predicting the electrical performance of electrically conductive metal materials. As Fig 1 illustrated, in the first step, the electrical performance data was obtained and subjected to data cleaning, followed by the partitioning of the original dataset into a training set and a test set. Subsequently, the raw data was normalized and standardized to mitigate the impact of varying data scales on the prediction results. In this study, we employed K-fold cross-validation as the method for data partitioning and model training, which has been proved as a well-established strategy [15–17]. K-fold cross-validation was utilized to reduce the influence of different data partitions on the training process. In this study, we set K to 5 for dataset splitting.The division ratio of 4:1 provides us with approximately 80% of the data for training, leaving 20% of the data for testing. Within the training data, a further 7:3 split is used to create a validation set, which is employed for assessing the model's performance and evaluating its ability to generalize to unseen data. Separating the validation set from the training set allows for a more robust

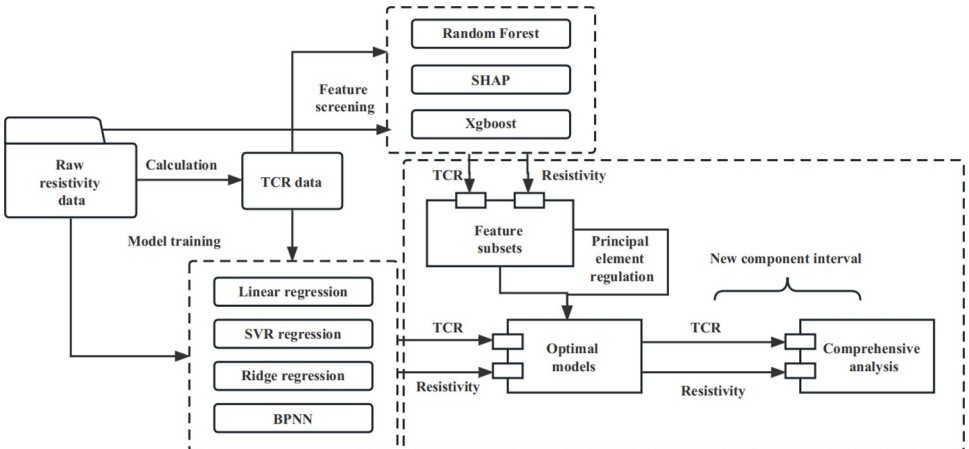

**Fig 1. Overview of the modeling process in simplified form.** After the optimal model is obtained, the variation trend of resistivity and TCR can be analyzed by adjusting the content of important elements based on the feature screening results.

estimation of the model's performance in practical applications. This balanced approach ensures that while a sufficient amount of data is retained for model performance evaluation, there is also ample training data to support the model's learning, thereby safeguarding its generalization capabilities.Machine learning methods exhibited varying sensitivities to material data within different ranges [18–21]. Hence, it was crucial to select proper algorithms based on specific material data samples and evaluate their performance using appropriate performance evaluation metrics. In this study, we evaluated the models using four metrics: root mean square error (RMSE), mean absolute error (MAE), mean absolute percentage error (MAPE), and coefficient of determination ($R^2$).Among these, $R^2$ serves as the principal performance metric to identify the most effective predictive model for subsequent analysis.

The second step involved feature importance selection. In this study, we utilized two different feature selection models and two evaluation methods to rank the importance of features. The intersection of the feature subsets obtained from both methods is selected as the final feature subset [22]. Since the feature data in this study comprised elemental composition data, the results of the feature importance selection represented the degree of influence that different elements have on the electrical performance of alloys. As our target performance included resistivity and TCR, separate feature selection was performed for each of these electrical performance indicators to identify the element subsets that have the greatest impacts on each of them.

The third step utilized the outcomes of the previous two steps for compositional optimization analysis. Due to the "trade-off" relationship between resistivity and TCR, achieving simultaneous positive gains in both electrical performance indicators cannot rely solely on blindly increasing or decreasing the content of certain elements based on feature selection results. In this study, by adjusting the element compositions obtained through feature selection, novel electrical performance data was calculated using the predictive model. The trends in resistivity and TCR changes were then comprehensively analyzed within the same range to explore the design principles for optimizing these two electrical performance indicators simultaneously. The details of each step will be discussed in the following subsections.

## Data source

The material electrical performance data used in this study primarily originated from the JMatPro database. The acquisition of material electrical performance data in this study

involved four main processes. Firstly, various types of alloy electrical performance data were collected through literature review, wherein the alloys were initially selected based on the values of resistivity and TCR. In this study, titanium alloys were chosen as the initial research objects. Next, programming was employed to generate 729 virtual sample points with titanium as the primary constituent element. These 729 titanium-based compositions included specific elements such as Al, Si, Zr, Mo, V, Sn, Nb, Mn, and Ti, totaling nine elements. Subsequently, the JMatPro high throughput modules (named API), which allows batch calculation, was utilized to obtain the resistivity data at 29 temperature points ranging from 25°C to 300°C for each virtual composition sample. A total of 21,141 resistivity data points were obtained for titanium alloy, constituting the resistivity dataset used in the following experiments. Lastly, by utilizing the calculation relationship between resistivity and TCR, a programming approach was employed to calculate the TCR data for the 729 compositions, forming the TCR dataset used in the following experiments.

## Calculation of TCR

With 25°C as the reference temperature, the relation between the resistance $R_t$ of the alloy at $t$ °C and the temperature can be expressed by the following formula:

$$R_t = R_{25}[1 + \alpha(t - 25) + \beta(t - 25)^2 + \gamma(t - 25)^3 + \ldots] \tag{1}$$

In Formula (1), $R_{25}$ is the resistance value at 25°C, and the unit is Ω; $\alpha$, $\beta$ and $\gamma$ are the resistance temperatures of 1st, 2nd and 3rd power respectively. Since the resistance value and temperature of titanium alloy in this paper are approximately linear in the range of 25°C ~300°C, the relationship between resistance and temperature can be simplified by Eq (1) as follows:

$$R_t = R_{25}[1 + \alpha(t - 25)] \tag{2}$$

From the relationship between resistance and resistivity, it can be seen that the change of resistance value of the same material at different temperatures essentially comes from the change of resistivity, so Formula (2) can be further simplified as:

$$\rho_t = \rho_{25}[1 + \alpha(t - 25)] \tag{3}$$

In Formula (3), $\rho_t$ and $\rho_{25}$ are the resistivity of titanium alloy at $t$°C and 25°C respectively, and the unit is Ω·m. As shown in Fig 2, $t$-25 is the x-axis and $\rho_t$ is the y-axis for linear fitting. The fitted intercept is $\rho_{25}$ and the slope is $\rho_{25}\alpha$, so the resistance temperature coefficient $\alpha$ can be calculated by dividing the slope by the intercept.

## Data normalization and standardization

Data normalization and standardization can also keep the original distribution of the data while eliminating the influence of the large difference in the order of magnitude of different features on the prediction model. Therefore, this study will normalize or standardize the original data before regression prediction and feature screening.

In normalization processing, each feature is transformed by the following formula:

$$x_i^* = \frac{x_i - \min(x)}{\max(x) - \min(x)} \tag{4}$$

Where: $x = (x_1, x_2, \cdots, x_m)$ represents the set consisting of the values of a feature of $m$ samples; $x_i \in x$; $\max(x)$ and $\min(x)$ are the maximum and minimum values in the feature set

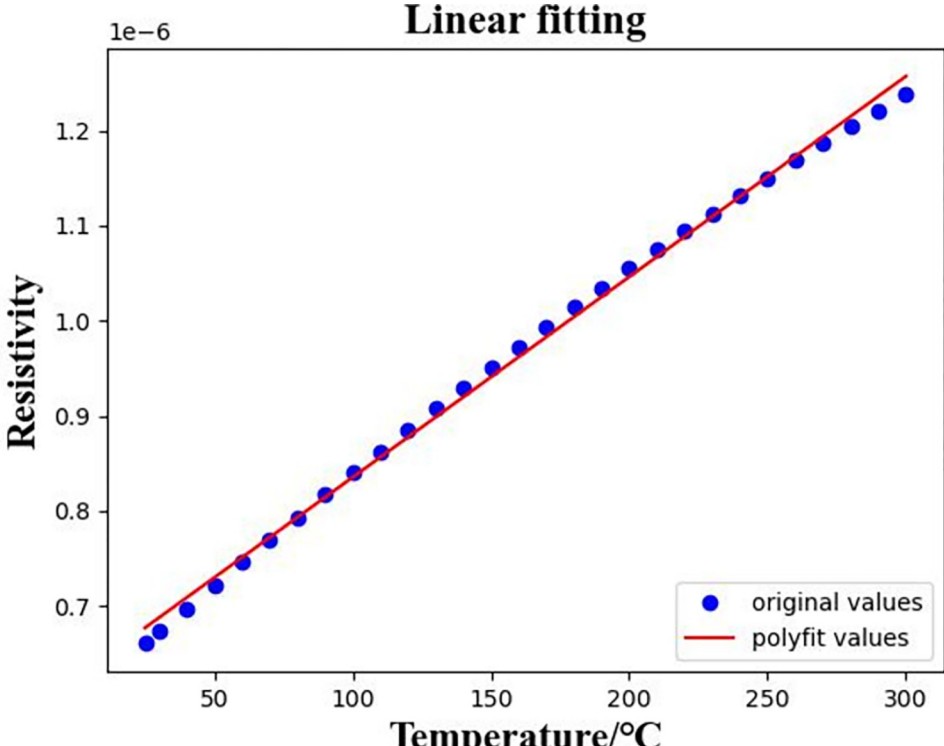

**Fig 2. Linear fitting curve of titanium alloy resistivity data for TCR calculation in the temperature range of 25˚C to 300˚C.**

respectively. $x_i^*$ indicates that the eigenvalue of $x_i$ in any value range is transformed into the value in the interval [0,1] through calculation, that is, the eigenvalue after normalization.

In normalization, each feature is transformed by the following formula:

$$x_i^* = \frac{x_i - mean(x)}{std(x)} \tag{5}$$

Where, $mean(x)$ and $std(x)$ are the characteristic mean and standard deviation respectively. The mean value of the data is 0 and the standard deviation is 1. The reason for the above data processing is that after comparing the two methods, it is found that the prediction effect after standardized processing is better.

## Machine learning regression model

In predictive tasks, machine learning can be divided into classification and regression [23–25]. The prediction of electrical properties in this paper is a regression task, wherein, four commonly-used machine learning models were employed: linear regression, ridge regression, support vector regression and BP neural network. Among them, BP neural network showed strong nonlinear mapping ability and was widely used in classification, fitting, diagnosis, prediction and other fields [26].

**Linear regression.** Linear regression model plays an important role in the field of machine learning, and its advantages are simple and easy to model. Its basic form is: Let the given data set $D = \{(x_1, y_1), (x_2, y_2), \cdots, (x_m, y_m)\}$, where $xi = (x_{i1}, x_{i2}, \cdots, x_{id})$, $m$ are the

number of samples, $d$ is the feature dimension, $y_i \in R$. Linear regression tries to learn:

$$f(xi) = \omega^T x_i + b \tag{6}$$

So that $f(x_i) \approx y_i$, where $\omega = (\omega_1, \omega_2, \cdots, \omega_d)$.The linear model has a certain interpretability. It can be seen from Eq (6) that $\omega_i$ represents the weight of each feature in the prediction, and $b$ is the target value when all features are zero. When using the least square method to solve $\omega$ and $b$, that is:

$$(\omega, b) = \arg \min \sum_{i=1}^{m} (y_i - \omega x_i - b)^2 \tag{7}$$

The optimal solution of $\omega$ and $b$ can be obtained from Eq (7).

**Ridge regression.**　Ridge regression is generally used in cases where the ratio of sample characteristic dimension to sample number is large, and it is a supplementary method to linear regression. The loss function of ridge regression model is:

$$\min_{\omega} \sum_{i=1}^{m} (y_i - \omega^T x_i)^2 + \lambda \|\omega\|_2^2 \tag{8}$$

L2 norm regularization is used in the expression, and the regularization parameter $\lambda > 0$.

Ridge regression can alleviate the overfitting problem by introducing regularization terms. In addition, from A mathematical point of view, Ridge regression can better solve the problem that linear regression can not get more stable $\omega$ or can not be solved.

**Support vector regression.**　The principle of support vector machine [27,28] is to find a hyperplane to divide different classes of samples. The objective function of the SVR problem is as follows:

$$\min \frac{1}{2} \|\omega\|^2 + C \sum_{i=1}^{m} l_\varepsilon (f(x_i) - y_i) \tag{9}$$

Where: $\omega = (\omega_1, \omega_2, \cdots, \omega_d)$ is the normal vector dividing the hyperplane; C is the regularization constant; $l_\varepsilon$ is the loss function.

$$l_\varepsilon = \begin{cases} 0, & |f(x) - y| \leq \varepsilon \\ |f(x) - y| - \varepsilon, & |f(x) - y| > \varepsilon \end{cases} \tag{10}$$

SVR can be expressed by the formula with kernel function as:

$$f(x) = \sum_{i=1}^{m} (\wedge \alpha_i - \alpha_i) \kappa(x, x_i) + b \tag{11}$$

Where, $\kappa(x, x_i)$ is the kernel function, $\wedge \alpha_i$ and $a_i$ are Lagrangian multipliers of the $i$th sample, $\wedge \alpha_i \geq 0$, $a_i \geq 0$. The SVR cores used in this study are Gaussian radial basis (RBF kernel) and S-type kernel (Sigmod).

**BP neural network.**　BP neural network is the extension of perceptron, the main characteristics of this network are the forward transmission of working signal and the reverse propagation of error signal. The internal network layer structure can be divided into three parts, namely, the input layer, the hidden layer and the output layer, as shown in Fig 3. Each layer of the network contains a number of neurons, and the layers are connected in a fully connected way, and the output of the upper layer is the input of the next layer, so as to realize the feature mapping and mining the inherent law of the data.

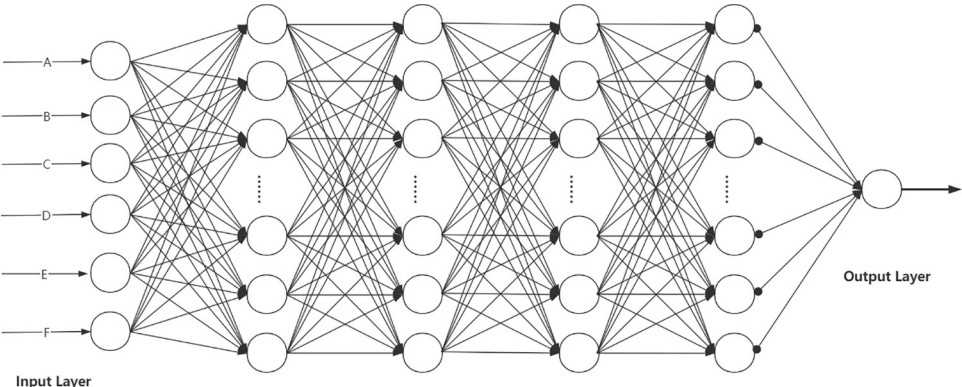

**Fig 3. BP neural network structure diagram.**

Suppose that the number of neurons in layer $l$ is $n$, then the output vector of layer $l$ is $a^l \in R^{n \times l}$. The output of layer $l$ can be expressed by the formula:

$$a^l = \sigma(z^l) = \sigma(W^l a^{l-1} + b^l) \tag{12}$$

Where: $z^l \in R^{n \times l}$ is the input vector of layer $l$; $W^l \in R^{n_{l-1} \times n_l}$ is the weight matrix from layer $l$-1 to layer $l$; $b^l \in R^{n \times 1}$ is the bias vector of the $l$ layer; $\sigma(\cdot)$ is the activation function. Common activation functions include Relu, Tanh, Sig-moid and so on.

**Evaluation index.** The root mean square error (RMSE), mean absolute error (MAE), mean absolute percentage error (MAPE) and goodness of Fit ($R^2$) were used to evaluate the performance of the prediction model.

RMSE is calculated as follows:

$$RMSE = \sqrt{\frac{1}{m} \sum_{i=1}^{m} (y_i - \hat{y}_i)^2} \tag{13}$$

MAE is calculated as follows:

$$MAE = \frac{1}{m} \sum_{i=1}^{m} |y_i - \hat{y}_i| \tag{14}$$

MAPE is calculated as follows:

$$MAPE = \frac{1}{m} \sum_{i=1}^{m} |\frac{y_i - \hat{y}_i}{y_i}| \tag{15}$$

The maximum value of $R^2$ is 1. The closer the value is to 1, the better the fitting degree is. The calculation method is as follows:

$$R^2 = 1 - \sum_{i=1}^{m} (y_i - \hat{y}_i)^2 / \sum_{i=1}^{m} (y_i - \bar{y})^2 \tag{16}$$

Among the above types, $y_i$ and $\hat{y}_i$ are the real and predicted values of the $i$ sample respectively, with the mean value $\hat{y} = 1$.

**Grid parameter optimization.** In the realm of machine learning, models typically comprise numerous hyperparameters that require careful tuning to achieve optimal performance.

These hyperparameters encompass factors such as learning rate, the number of trees, depth, and regularization strength. The selection of the right combination of hyperparameters is pivotal for attaining high-performance models.

The fundamental principle of grid search involves exploring a predefined grid of hyperparameter value combinations. Initially, a grid of hyperparameter combinations is constructed based on the hyperparameters and their candidate values. It is essential to exercise caution when choosing hyperparameters and candidate values to prevent the search space from becoming excessively large, consuming excessive computational resources. Subsequently, for each hyperparameter combination, models are trained on the training data, and their performance is evaluated using techniques such as cross-validation. Finally, based on the results of performance metrics, the hyperparameter combination exhibiting the best performance is selected. This combination typically exhibits the lowest error or the highest accuracy.

This study necessitates the use of three models for grid search hyperparameter optimization: BPNN, Support Vector Machine, and Ridge Regression. The grid parameter settings for these three models are as follows:

1.BPNN Grid Parameter Settings:

- Hidden layer sizes setting: [(36,), (50,), (100,)]

- Activation function setting: ['relu', 'tanh']

- Optimizer setting: ['adam', 'lbfgs']

- Maximum iteration setting: [100000, 200000]

2.Support Vector Machine Grid Parameter Settings:

- Regularization coefficient C setting: [0.01, 0.1, 1, 10, 50, 100, 150]

- Epsilon tolerance setting: [0.01, 0.1, 1, 10]

- Kernel function coefficient gamma setting: [0.01, 0.1, 1, 10]

3.Ridge Regression Grid Parameter Settings:

- Regularization parameter alpha setting: [0.001, 0.01, 0.1, 1, 10, 100]

## Feature selection

In traditional regression analysis models, the more features means the less accuracy the predictions become. Therefore, feature selection is necessary to improve results and optimization.

Feature engineering is a crucial step in machine learning [29–31], and different feature choices can significantly impact the prediction outcomes. In the context of titanium alloy electrical performance prediction, when the prediction model utilized elemental features to forecast target electrical performance, each elemental feature became a factor influencing the prediction results. The task of feature selection is to identify highly correlated features with the target performance while filtering out irrelevant features, thereby improving prediction effectiveness and reducing computational complexity. The commonly used feature selection methods mainly include three types: filter-based, embedded, and wrapper-based approaches [32]. In this study, a wrapper-based feature selection method was employed, which combined feature subsets with machine learning models and utilized the predictive performance of the models as the criterion for selecting the feature subsets. Since the feature selection process involves multiple iterations of training the learner, wrapper-based feature selection entails

significant computational complexity and time consumption. This paper employed two wrapper-based feature selection algorithms, namely random forest and Xgboost, with feature importance and SHAP (SHapley Additive exPlanations) values used to illustrate the feature selection results.

**Random forest feature selection.** Random forest adopts random feature selection method to reduce the overfitting risk and increase the generalization ability of the model by randomly selecting different feature subsets during the construction of each decision tree.

The random forest defines the importance measure of $X$ feature. The steps to calculate the importance of a feature are as follows:

1. The prediction Error rate of random forest for samples outside the bag is called Out-Of-Bag error (OOB). For decision tree $T_i$ in random forest, the number of classification errors $E_i$ on the data outside the bag is calculated.

2. The value of $X$ is randomly disturbed in the OOB data of the decision tree, and the number of classification errors $E_i^X$ is recalculated.

3. Let $i = 1,2,\ldots,n$, repeat the above two steps, where $n$ is the number of decision trees contained in the random forest.

4. The importance of feature $X$ is defined as:

$$Ix = \frac{1}{n}\sum_{i=1}^{n}(E_i^X - E_i) \tag{17}$$

The basis of this definition is that if the OOB error of the model is significantly increased after the addition of noise to a feature, it indicates that the feature has a greater impact on the prediction result, and thus has a higher importance.

**Xgboost feature selection.** Xgboost uses incremental training and fine-tuning of feature split points, so that variable selection and weight parameter adjustment can be optimized in continuous iteration to improve model performance. The core idea of the Xgboost algorithm is to generate trees by constantly splitting features, and for each tree generated, a new function is generated to fit the residual of the last prediction. When we need to predict a sample, each sample will fall on the corresponding leaf node in each tree, and will correspond to a score, and finally add the scores corresponding to each tree to get the predicted value of the sample.

The objective function of Xgboost algorithm is:

$$Obj^{(t)} = \sum_{i=1}^{n} l(y_i, \hat{y}_i^{(t-1)} + f_t(x_i)) + \Omega(f_t) + const \tag{18}$$

In Eq (18), $l$ is the loss function, $\Omega(f_t)$ is the regular term, and *const* is the constant term.

**SHAP.** SHAP is a widely recognized attribution method in the field of explainable machine learning, derived from the Shapley value in cooperative game theory [33]. The key to interpreting machine learning models using SHAP analysis is to calculate the Shapley value for each feature of every instance. In practical applications, the Shapley value is approximated, and SHAP is an optimization algorithm for estimating Shapley values.

This method possesses three desirable properties of Shapley value theory: local accuracy, missingness, and consistency. Local accuracy refers to the sum of feature attributions being equal to the difference between the expected output and the actual output of the explanatory model, enabling the explanatory model to capture the discrepancy between the desired output and the observed output of a given instance [34]. Missingness implies that the Shapley value for a feature that is absent is considered as 0, indicating no contribution to the prediction.

Consistency means that if a change in the model leads to an increase or unchanged marginal contribution of a particular feature, the Shapley value for that feature will not decrease. In addition to these features, the greatest advantage of SHAP lies in its independence from the predictive model [35], allowing for the theoretical interpretation of any machine learning model.

## Results and analysis

The target performance data and feature data for the experiment were obtained by fitting and calculating 21141 titanium alloy resistivity samples using Eq (3), resulting in 729 sets of TCR (Temperature Coefficient of Resistance) data. The resistivity data at 25˚C for the 729 sets of titanium alloys, as well as the composition proportion data containing nine elements, constituted the experimental feature data. The target performance data and feature data together formed the experimental dataset, which is summarized in Table 1. And the statistical analysis results for Resistivity and TCR are individually presented in (a) and (b) of Fig 4.

In Table 1, "Count" represents the total number of data points for each data type, which is 729 sets in this case. "Mean" denotes the average value of each data type, while "Std" represents the standard deviation. "Min" and "Max" correspond to the minimum and maximum values, respectively. For elemental compositions, they also indicate the compositional range of each element. It is evident that the titanium (Ti) content, as the base element for titanium alloys, ranges from 66.28% to 98.53%, making it the most abundant element among all. In Fig 4(A), the histogram represents the distribution of resistivity values, with the vertical axis indicating the frequency of resistivity values. The purple curve represents the kernel density estimation curve. From the frequency histogram, it is evident that the primary range of resistivity values falls between 0.7 and 1.6. The kernel density estimation curve exhibits three peaks, corresponding to intervals with a higher concentration of data points, namely around 0.9, 1.2, and 1.4. In Fig 4(B), the histogram depicts the distribution of TCR (Temperature Coefficient of Resistance) values, with the vertical axis showing the frequency of TCR values. The purple curve represents the kernel density estimation curve. The frequency histogram reveals that the main range of TCR values lies between 500 and 3000. The kernel density estimation curve indicates a gradual decrease in the number of samples with high TCR values. This reduction can be attributed to the fact that, in existing alloy materials, obtaining high TCR values is relatively more challenging compared to high resistivity values.

After normalizing or standardizing the data in Table 1 through preprocessing operations, subsequent prediction modeling, feature selection, and optimization range analysis were performed using the resistivity and TCR as target performance data.

### Results of model prediction

#### Predicted results of resistivity model

Table 2 presents the predicted results of resistivity data based on five machine learning models, and the values of four evaluation metrics are based on the average level of K-fold cross-

**Table 1. Training dataset description for predictive modeling: TCR, resistivity values, and concentrations of nine elemental characteristics.**

|       | $\rho_{25}$ | TCR | Al | Si | Zr | Mo | V | Sn | Nb | Mn | Ti |
|-------|------|---------|------|------|------|------|------|------|-------|------|-------|
| Count | 729 | 729 | 729 | 729 | 729 | 729 | 729 | 729 | 729 | 729 | 729 |
| Mean | 1.16 | 1284.18 | 2.56 | 0.08 | 3.31 | 0.86 | 1.50 | 1.25 | 6.63 | 1.40 | 82.40 |
| Std | 0.24 | 670.93 | 2.09 | 0.00 | 2.71 | 0.71 | 1.23 | 1.02 | 5.41 | 1.60 | 6.64 |
| Min | 0.66 | 348.81 | 0.00 | 0.08 | 0.00 | 0.00 | 0.00 | 0.00 | 0.00 | 1.40 | 66.28 |
| Max | 1.62 | 3194.46 | 5.13 | 0.08 | 6.63 | 1.75 | 3.00 | 2.50 | 13.25 | 1.40 | 98.53 |

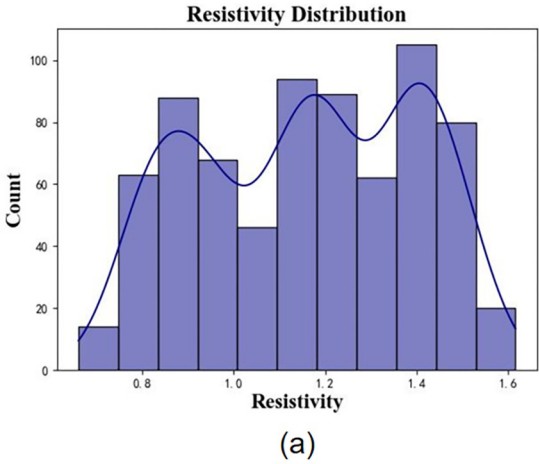

(a)

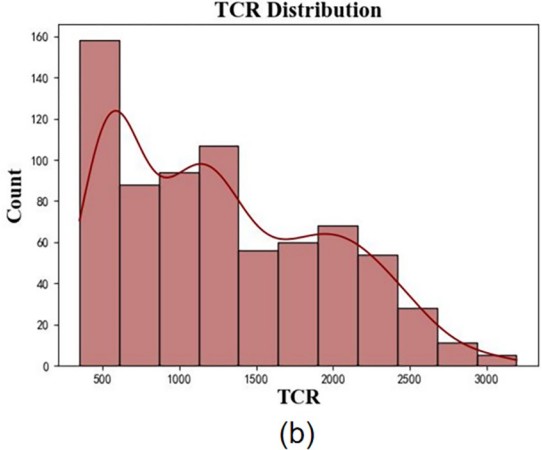

(b)

**Fig 4. Statistical analysis histograms for resistivity and TCR.** (a) presents the statistical analysis results for Resistivity;(b) displays the statistical analysis results for TCR.

validation. The BPNN (Backpropagation Neural Network) model and three SVR (Support Vector Regression) models with different kernel functions are displayed with their best results after parameters tuning.

From the results shown in Table 2, it can be observed that the Rbf (Radial Basis Function) kernel SVR model performs the best among all models, with superior results in all four evaluation metrics, and the coefficient of determination R2 exceeding 0.99.

The parameters of the Rbf-SVR model for resistivity prediction are as follows: The Rbf kernel is chosen for SVR, with a regularization coefficient C of 150, a kernel coefficient gamma of 0.1, and ε set to 0.01.

**Table 2. Resistivity prediction results from 5 different models.**

| Resistivity Prediction Models | MAE | MAPE | RMSE | $R^2$ |
|---|---|---|---|---|
| BPNN | 0.026 | 2.48% | 0.034 | 0.981 |
| Linear | 0.028 | 2.68% | 0.037 | 0.978 |
| Ridge | 0.028 | 2.68% | 0.037 | 0.978 |
| Rbf-SVR | 0.014 | 1.35% | 0.018 | 0.995 |
| Sigmod-SVR | 0.031 | 2.98% | 0.044 | 0.969 |

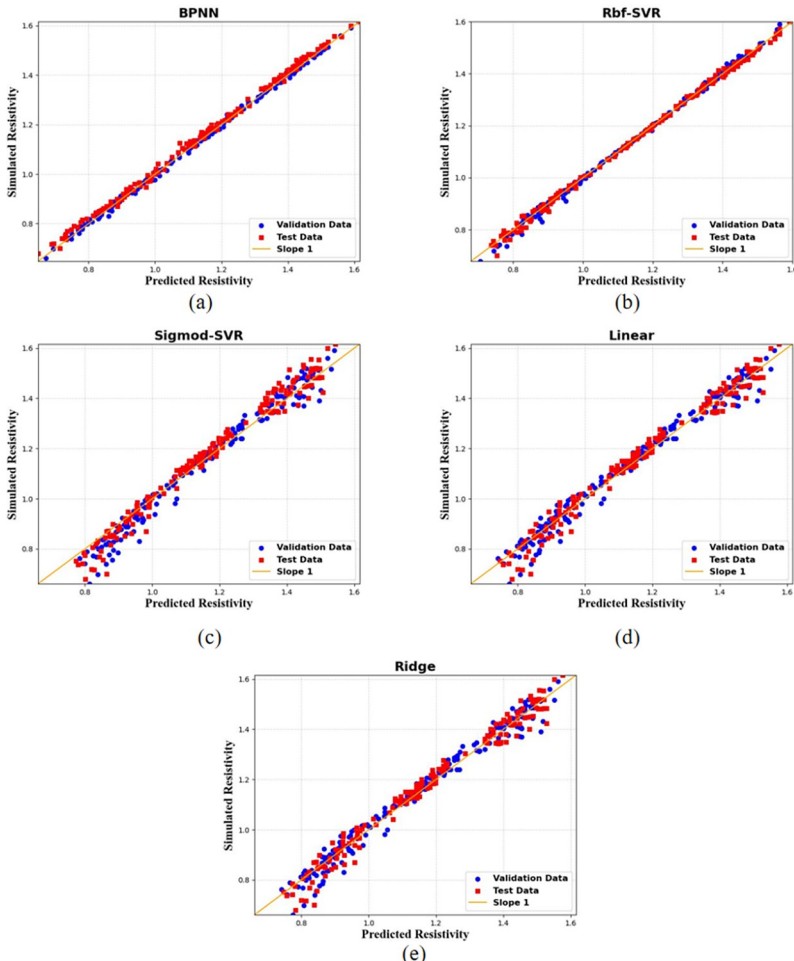

**Fig 5. Cross-plots of simulated and predicted resistivity values for five machine learning models evaluated on training and testing subsets.** Panels (a) through (e) correspond to the BPNN model, Rbf-SVR model, Sigmod-SVR model, linear regression model, and ridge regression model, respectively.

In Fig 5(A) to 5(E), the regression performance of five machine learning models on the validation and test datasets is presented. It is evident that the SVR model based on the Rbf kernel demonstrates the best regression performance, with minimal discrepancies between the predicted and actual resistivity values, resulting in an overall excellent predictive performance. In conclusion, the Rbf-SVR model is selected as the predictive model for resistivity, the target property in this study.

## Prediction results of TCR model

Table 3 presents the prediction results of TCR data based on five machine learning models, and the values of the four evaluation metrics are still based on the average level of K-fold cross-validation. Similarly, the BPNN model and three SVR models with different kernel functions are displayed with their best results obtained during parameter tuning.

From the results shown in Table 3, it can be observed that the BPNN model achieves the best prediction performance among all models, with superior results in all four evaluation metrics, especially with a coefficient of determination R2 exceeding 0.99. The Rbf kernel SVR model follows closely in terms of prediction performance, but the errors, as indicated by MAE

**Table 3. TCR prediction results from 5 different models.**

| TCR Prediction Models | MAE | MAPE | RMSE | $R^2$ |
|---|---|---|---|---|
| BPNN | 27.32 | 2.29% | 42.08 | 0.996 |
| Linear | 118.03 | 15.91% | 154.80 | 0.951 |
| Ridge | 117.96 | 15.88% | 154.78 | 0.951 |
| Rbf-SVR | 55.82 | 4.61% | 93.31 | 0.981 |
| Sigmod-SVR | 117.73 | 10.11% | 182.99 | 0.932 |

and MAPE, are approximately twice that of the BPNN model. The linear regression, ridge regression, and Sigmod kernel SVR models exhibit similar prediction performance, however, their performance is significantly inferior compared to the BPNN model.

The parameters of the BPNN model for TCR prediction are as follows: The neural network has two hidden layers, each with 36 nodes, and the activation function is set as Relu. The weight optimizer is set to stochastic gradient descent, with a regularization parameter of 0.0001, and the maximum number of iterations is set to 10,000.

In Fig 6(A) to 6(E), the regression performance of five machine learning models on the validation and test datasets is depicted. It is evident that the BPNN model demonstrates the best

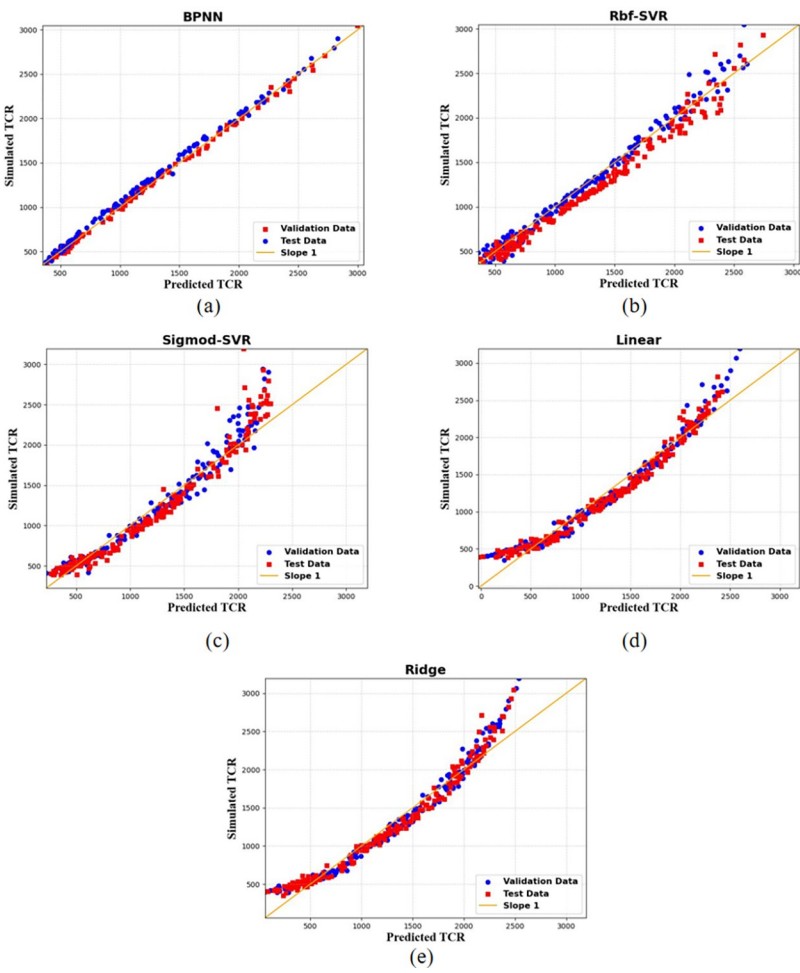

**Fig 6. Cross-plots of simulated and predicted TCR values for five machine learning models evaluated on training and testing subsets.** Panels (a) through (e) correspond to the BPNN model, Rbf-SVR model, Sigmod-SVR model, linear regression model, and ridge regression model, respectively.

regression performance, with minimal discrepancies between the predicted and actual TCR values, resulting in an overall excellent predictive performance. In conclusion, the BPNN model is selected as the predictive model for TCR, the target property in this study.

## Feature selection

### Resistivity feature selection

Since the Ti element is the base element and its content often varies with the content of other trace elements. As a result, Ti is not involved in the feature selection process. The Si and Mn elements are trace elements with fixed composition, so these two elements are also excluded from the feature selection process. The impact of different elements on the resistivity target performance was investigated using two different machine learning models, Random Forest and Xgboost, and the results are shown in Fig 7.

Fig 7(A) and 7(B) present the feature selection results based on feature importance. From Fig 6(A), it can be observed that the top three elements ranked by feature importance using the Random Forest model are Al, Zr, and Nb. On the other hand, Fig 7(B) shows that the top three elements based on feature importance using the Xgboost model are Al, Zr, and Sn. Therefore, the feature selection for resistivity includes the common elements of feature importance from both machine learning methods: Al and Zr.

In Fig 7(C) and 7(D) depict feature contribution representations based on SHAP values. In comparison to the feature importance method, which directly provides the importance ratio, the SHAP approach also reveals the polarity of the impact. The SHAP visualization results in (c) and (d) show close similarities. In the SHAP plots, each row represents a feature, with the x-axis indicating the SHAP values. The features are sorted based on the mean absolute value of

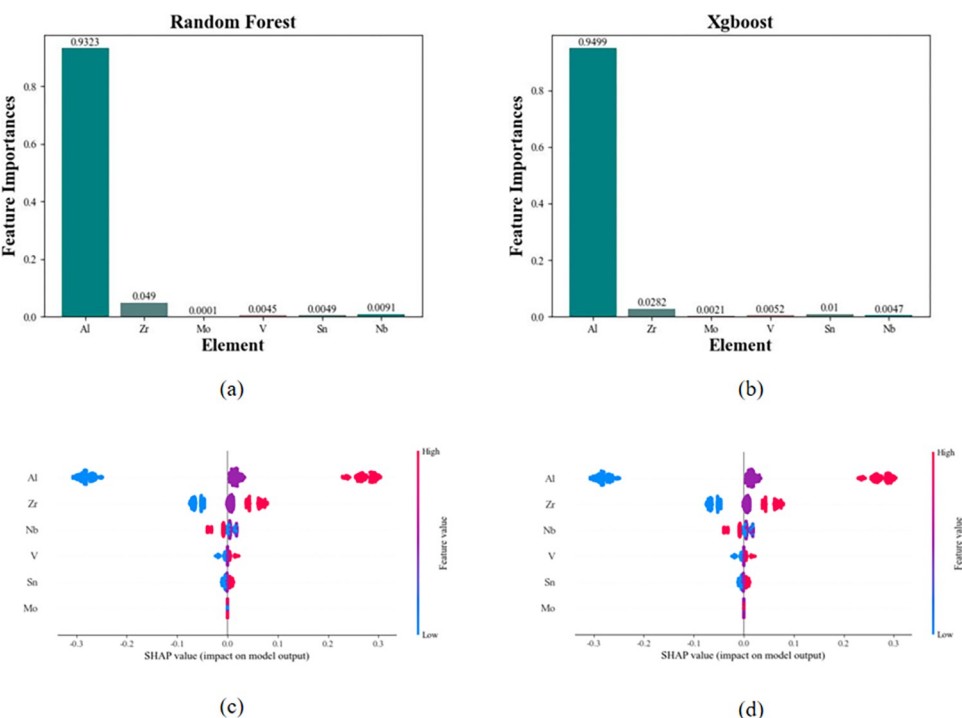

**Fig 7. Characteristic screening results of resistivity.** (a) and (b) represent the resistivity feature importance results based on random forest and Xgboost machine learning methods, respectively; (c) and (d) respectively represent the results of the two methods on SHAP.

SHAP, highlighting the most significant features for the model. Wider regions signify a concentration of samples, where each point represents an individual sample. Warmer colors, such as red, indicate larger feature values, while cooler colors, like blue, correspond to smaller feature values.

It is evident that the content of the Al element is highly significant for the model. Samples with higher Al element content, represented by red points with high SHAP values (greater than 0), exert a positive influence. Conversely, for the blue-shaded portion where feature values are smaller, SHAP values are negative (less than 0), indicating a negative influence. From a horizontal perspective, the distribution of samples for the Al element feature is more scattered, signifying a greater impact of this feature. Additionally, for features like Mo element, most data points are dispersed around SHAP = 0, suggesting that the Mo element has a minimal impact within the normal range.

Through the SHAP plots, it becomes apparent that both Al and Zr have a similar impact on resistivity. Specifically, within a certain range, higher concentrations of Al and Zr lead to increased resistivity. This insight provides valuable guidance for subsequent compositional optimization analysis.

**TCR feature selection.** Similarly, Ti, Si, and Mn are not involved in the feature selection process for TCR. The impact of different elements on the TCR target performance was investigated using the Random Forest and Xgboost models, and the results are shown in Fig 8.

Fig 8(A) and 8(B) present the feature selection results based on feature importance. From Fig 8(A), it can be observed that the top three elements ranked by feature importance using the Random Forest model are Al, Zr, and V. Similarly, Fig 8(B) shows that the top three elements based on feature importance using the Xgboost model are Al, Zr, and V. Therefore, the

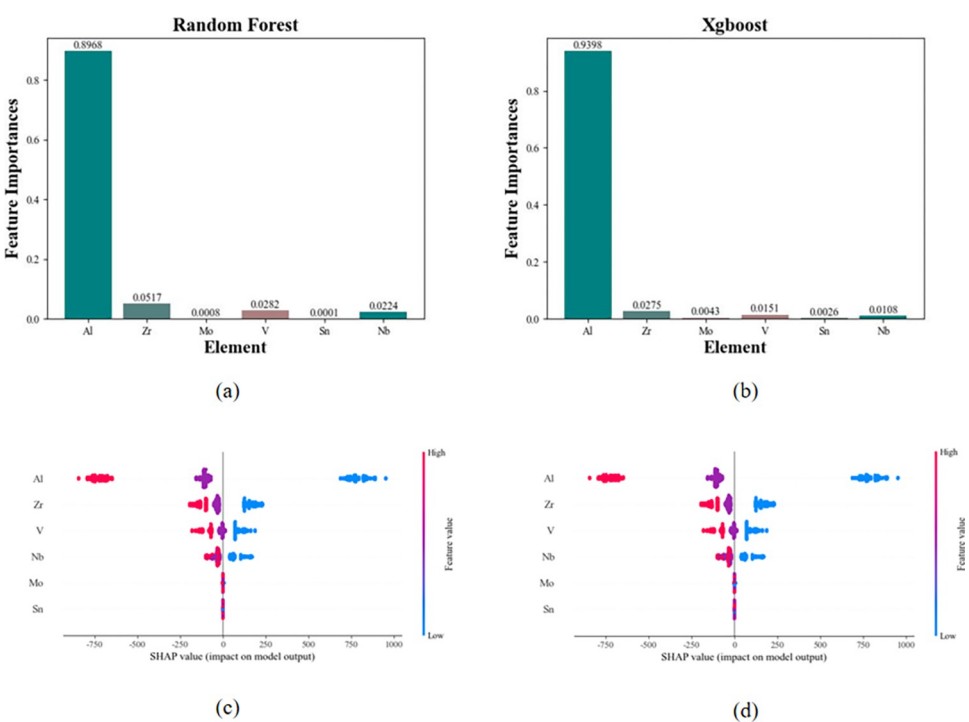

**Fig 8. Characteristic screening results of CTR.** (a) and (b) represent the TCR feature importance results based on random forest and Xgboost machine learning methods; (c) and (d) represent the results of the two methods on SHAP.

feature selection for TCR includes the common elements of feature importance from both machine learning methods: Al, Zr, and V.

Fig 8(C) and 8(D) depict the feature contributions based on SHAP values. It can be observed that the impacts of Al, Zr, and V on TCR are opposite to that on resistivity. Within a certain range, increasing the content of Al, Zr, and V leads to decrease in TCR, which is consistent with the reciprocal relationship between resistivity and TCR during compositional design.

## Composition optimization analysis

Based on the feature selection results for resistivity and TCR, the variable elements chosen for compositional optimization are Al and Zr. When adjusting the content of Al and Zr, the content of other elements is set to the average values, as listed in Table 1. The newly generated virtual samples are used as the test set, and the resistivity is predicted using the SVR model with the Rbf kernel and the TCR is predicted using the BPNN model with the optimal parameters. The trends of resistivity and TCR variations are then analyzed in the same coordinate system.

**Effect of Al on electrical properties.** The average values for Si, Zr, Mo, V, Sn, Nb, and Mn are set to 7.50%, 3.31%, 0.86%, 1.50%, 1.25%, 6.63%, and 1.40% respectively. According to Table 1, the composition range of Al is set to 0–5% with a step size of 0.1. The content of Ti changes with the content of Al. In total, 51 virtual samples are generated.

These 51 virtual samples are the test set for prediction using the Rbf-SVR and BPNN models. The resulting curves of resistivity and TCR prediction data are shown in Fig 9. The red curve represents the resistivity variation caused by the change in Al content from 0% to 5%, while the green curve represents the TCR variation. It can be observed that as resistivity increases, TCR decreases significantly. Therefore, when designing the target electrical performance of titanium alloys, it is important to consider both resistivity and TCR variations. The yellow dashed line in Fig 9. corresponds to an Al content of 3.8%. The intersection of the dashed line with the two curves represents the values of both electrical properties. It can be seen that at this point, the resistivity performance is good, but the TCR is low. To improve the TCR performance of the titanium alloy, the dashed line needs to shift to the left, indicating a

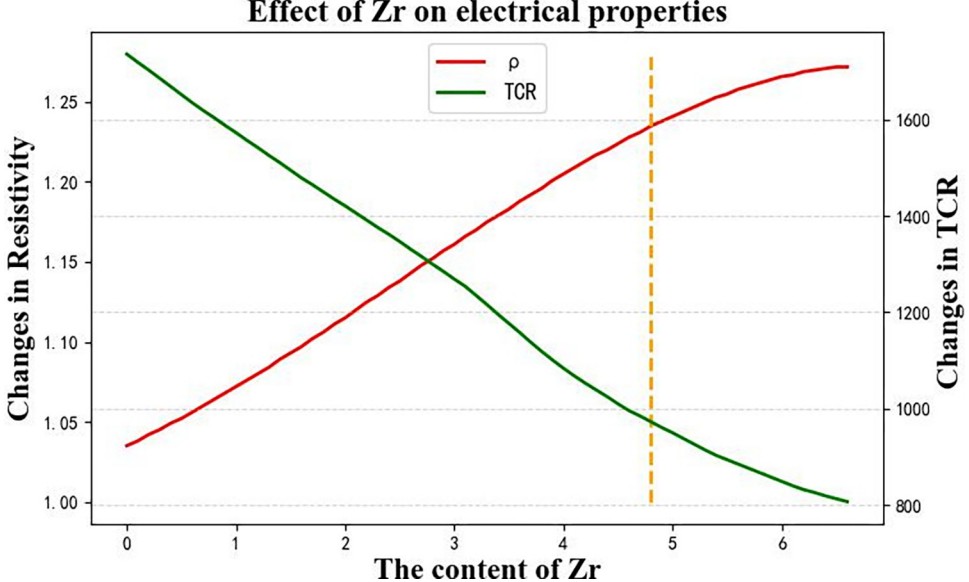

**Fig 9. Effect of Al content change on electrical properties of titanium alloy.**

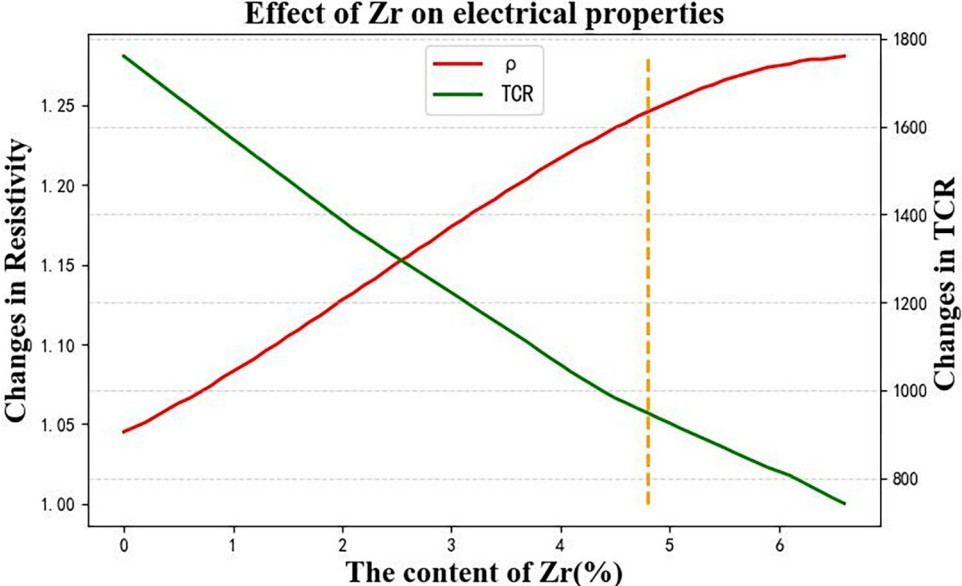

**Fig 10. Effect of Al content change on electrical properties of titanium alloy.**

decrease in Al content. Based on this optimization approach, it is evident that the intersection points of the two curves correspond to satisfactory levels of both resistivity and TCR. In this study, both resistivity and TCR are targeted for high performance. Therefore, the range of Al content can be considered within the vicinity of the intersection points shown in Fig 9, approximately around 1.5% to 2%.

**Effect of Zr on electrical properties.** The average values for Al, Si, Mo, V, Sn, Nb, and Mn are set to 2.56%, 7.50%, 0.86%, 1.50%, 1.25%, 6.63%, and 1.40% respectively. According to Table 1, the composition range of Zr is set to 0–6.6% with a step size of 0.1. The content of Ti changes with the content of Zr. In total, 67 virtual samples are generated.

These 67 virtual samples are the test set for prediction using the Rbf-SVR and BPNN models. The resulting curves of resistivity and TCR prediction data are shown in Fig 10. The red curve represents the resistivity variation caused by the change in Zr content from 0% to 6.6%, while the green curve represents the TCR variation. The yellow dashed line in Fig 10 corresponds to a Zr content of 4.8%, where the resistivity performance is good and the TCR is low. Therefore, when designing the Zr content for the titanium alloy, the range can be considered around the intersection points shown in Fig 10, approximately around 2.5% to 3%.

If the target performance leans towards either resistivity or TCR, this optimization approach can still be used for design purposes.

## Conclusions

1. The support vector machine model with the radial basis function kernel, Rbf-SVR, achieved the best performance in predicting the resistivity of titanium alloys, with a correlation coefficient of 0.995. The absolute percentage error between the true values and predicted values of the test samples is within 2%. The backpropagation neural network model with two hidden layers, BPNN, performed the best in predicting the TCR of titanium alloys, with a correlation coefficient of 0.996. The absolute percentage error between the true values and

predicted values of the test samples is within 3%. Both prediction models exhibited high accuracy and good generalization ability.

2. Feature selection was performed using two different machine learning models, Random Forest and Xgboost, for resistivity and TCR. The feature importance intersection elements for resistivity are Al and Zr, with a positive effect on resistivity. The feature importance intersection elements for TCR are Al, Zr, and V, with a negative effect on TCR.

3. For titanium alloy thermoelectric material, the electrical performance of the alloy can be improved by adjusting the content of Al and Zr. However, considering the trade-off relationship between resistivity and TCR, to achieve higher gains in both resistivity and TCR, it is recommended to set the composition range of Al at around 1.5% to 2% and the composition range of Zr at around 2.5% to 3%.

## Supporting information

**S1 Data.**
(XLSX)

## Author Contributions

**Resources:** Qiang Peng.

**Validation:** Degang Xu.

**Writing – original draft:** Chengqun Zhou, Muyang Pei.

**Writing – review & editing:** Chao Wu, Guoai He.

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
