## [Decision Letter · Decision Letter 0]

10 Oct 2023

PONE-D-23-26687Development of new materials for electrothermal metals using data driven and machine learningPLOS ONE

Dear Dr. He,

Thank you for submitting your manuscript to PLOS ONE. After careful consideration, we feel that it has merit but does not fully meet PLOS ONE’s publication criteria as it currently stands. Therefore, we invite you to submit a revised version of the manuscript that addresses the points raised during the review process.

Reviewer #1

After conducting a thorough review of the manuscript, I would like to provide the following comments:

1. The abstract should include the specific names of the machine learning methods used in this study.

2. The method employed to adjust the control parameters in the neural network is clearly stated and analyzed. It is recommended to provide a similar level of detail for other algorithms utilized in this manuscript.

3. To enhance the machine learning section, I suggest considering the following articles as additional references:

- Drilling rate prediction from petrophysical logs and mud logging data using optimized Multi Layer Perceptron neural network

- Application of hybrid artificial neural networks for predicting rate of penetration (ROP): A case study from Marun oil field

- A machine learning approach to predict drilling rate using petrophysical and mud logging data

- A geomechanical approach to casing collapse prediction in oil and gas wells aided by machine learning

- Developing a new rigorous drilling rate prediction model using a machine learning technique

- Geomechanical parameter estimation from mechanical specific energy using artificial intelligence

- Prediction of permeability from well logs using a new hybrid machine learning algorithm

- Hybrid machine learning algorithms to enhance lost-circulation prediction and management in the Marun oil field

- Optimization of controllable drilling parameters using a novel geomechanics-based workflow

- Shear wave velocity prediction from petrophysical logs using MLP-PSO algorithm

- A new approach to mechanical brittleness index modeling based on conventional well logs using hybrid algorithms

4. It is important to clarify how the separation ratio for training and test data was determined. Please explain the rationale behind choosing this specific ratio.

5. In the captions of tables and figures, it should be explicitly specified for which category of data they are intended.

6. For further analysis, I recommend considering the following articles as valuable references:

- Combined machine-learning and optimization models for predicting carbon dioxide trapping indexes in deep geological formations

- Predicting uniaxial compressive strength from drilling variables aided by hybrid machine learning

- Hybridized machine-learning for prompt prediction of rheology and filtration properties of water-based drilling fluids

- Machine-learning models to predict hydrogen uptake of porous carbon materials from influential variables

- Machine-learning predictions of solubility and residual trapping indexes of carbon dioxide from global geological storage sites

7. Among the various criteria considered, it is essential to mention in the text which specific criterion was used to select the best model.

8. It would be beneficial to provide additional explanations regarding the figures related to SHAP (Shapley Additive exPlanations). Please elaborate on the specific details and insights that can be derived from these figures.

Reviewer #2

Please find the review below:

1) Statistical analysis no present or weak lacks plots (histograms, heat map etc)

2) Entire work is not novel. Why is ML used for this, simply adding ML to existing work has no new science or value

3) Entire document needs to be rewritten, introduction lack's structure (research question, the knowledge gap and how this method is only one to fill that gap)

4)TCR normalized per unit area plotted against length is needed

5) Not sure why Al and Zr was choosen, did you measure composition changes of device using XPS/EDX for materials analysis, need proofs for that

6) Very weak electro thermal tasting analysis requires more results (for ex R vs temperature plots, thermal conductivity measurements etc.)

7) Figures are not presented in proper way and looks like a master's thesis document or Lab manual

We look forward to receiving your revised manuscript.

Kind regards,

Babatunde Abiodun Salami

Academic Editor

PLOS ONE

Journal Requirements:

This work was Funded by Shenzhen Zhuolineng Technology Co., Ltd; National Natural Science Foundation of Hunan province (2022JJ30721).

Reviewers' comments:

Reviewer's Responses to Questions

**Comments to the Author**

1. Is the manuscript technically sound, and do the data support the conclusions?

Reviewer #1: Yes

Reviewer #2: No

2. Has the statistical analysis been performed appropriately and rigorously? 

Reviewer #1: Yes

Reviewer #2: No

3. Have the authors made all data underlying the findings in their manuscript fully available?

Reviewer #1: Yes

Reviewer #2: Yes

4. Is the manuscript presented in an intelligible fashion and written in standard English?

Reviewer #1: Yes

Reviewer #2: No

5. Review Comments to the Author

Reviewer #1: After conducting a thorough review of the manuscript, I would like to provide the following comments:

1. The abstract should include the specific names of the machine learning methods used in this study.

2. The method employed to adjust the control parameters in the neural network is clearly stated and analyzed. It is recommended to provide a similar level of detail for other algorithms utilized in this manuscript.

3. To enhance the machine learning section, I suggest considering the following articles as additional references:

- Drilling rate prediction from petrophysical logs and mud logging data using optimized Multi Layer Perceptron neural network

- Application of hybrid artificial neural networks for predicting rate of penetration (ROP): A case study from Marun oil field

- A machine learning approach to predict drilling rate using petrophysical and mud logging data

- A geomechanical approach to casing collapse prediction in oil and gas wells aided by machine learning

- Developing a new rigorous drilling rate prediction model using a machine learning technique

- Geomechanical parameter estimation from mechanical specific energy using artificial intelligence

- Prediction of permeability from well logs using a new hybrid machine learning algorithm

- Hybrid machine learning algorithms to enhance lost-circulation prediction and management in the Marun oil field

- Optimization of controllable drilling parameters using a novel geomechanics-based workflow

- Shear wave velocity prediction from petrophysical logs using MLP-PSO algorithm

- A new approach to mechanical brittleness index modeling based on conventional well logs using hybrid algorithms

4. It is important to clarify how the separation ratio for training and test data was determined. Please explain the rationale behind choosing this specific ratio.

5. In the captions of tables and figures, it should be explicitly specified for which category of data they are intended.

6. For further analysis, I recommend considering the following articles as valuable references:

- Combined machine-learning and optimization models for predicting carbon dioxide trapping indexes in deep geological formations

- Predicting uniaxial compressive strength from drilling variables aided by hybrid machine learning

- Hybridized machine-learning for prompt prediction of rheology and filtration properties of water-based drilling fluids

- Machine-learning models to predict hydrogen uptake of porous carbon materials from influential variables

- Machine-learning predictions of solubility and residual trapping indexes of carbon dioxide from global geological storage sites

7. Among the various criteria considered, it is essential to mention in the text which specific criterion was used to select the best model.

8. It would be beneficial to provide additional explanations regarding the figures related to SHAP (Shapley Additive exPlanations). Please elaborate on the specific details and insights that can be derived from these figures.

Reviewer #2: Please find the review below:

1) Statistical analysis no present or weak lacks plots (histograms, heat map etc)

2) Entire work is not novel. Why is ML used for this, simply adding ML to existing work has no new science or value

3) Entire document needs to be rewritten, introduction lack's structure (research question, the knowledge gap and how this method is only one to fill that gap)

4)TCR normalized per unit area plotted against length is needed

5) Not sure why Al and Zr was choosen, did you measure composition changes of device using XPS/EDX for materials analysis, need proofs for that

6) Very weak electro thermal tasting analysis requires more results (for ex R vs temperature plots, thermal conductivity measurements etc.)

7) Figures are not presented in proper way and looks like a master's thesis document or Lab manual

6. PLOS authors have the option to publish the peer review history of their article (what does this mean?). If published, this will include your full peer review and any attached files.

Reviewer #1: **Yes: **Mohammad Mehrad

Reviewer #2: No

---

## [Author Response · Author response to Decision Letter 0]

5 Nov 2023

Reviewer #1:

After conducting a thorough review of the manuscript, I would like to provide the following comments:

1. The abstract should include the specific names of the machine learning methods used in this study.

Response: Thank you for your careful review. In accordance with the reviewer's suggestions, we have now included the specific names of the machine learning methods used in the abstract.

2. The method employed to adjust the control parameters in the neural network is clearly stated and analyzed. It is recommended to provide a similar level of detail for other algorithms utilized in this manuscript.

Response: Thank you for the valuable suggestions from the reviewers. The parameter optimization method employed in this study is grid parameter optimization. In order to provide details on the optimization of these model parameters, we have included a dedicated subsection in Chapter 2 to specifically describe the configuration of grid parameters. We hope this will provide readers with a clearer understanding of the process of selecting the best parameters.

3. To enhance the machine learning section, I suggest considering the following articles as additional references:

Response: Following the recommendations of the reviewer, we have incorporated additional references related to machine learning in the introduction and methodology sections.

- Drilling rate prediction from petrophysical logs and mud logging data using optimized Multi Layer Perceptron neural network

- Application of hybrid artificial neural networks for predicting rate of penetration (ROP): A case study from Marun oil field

- A machine learning approach to predict drilling rate using petrophysical and mud logging data

- A geomechanical approach to casing collapse prediction in oil and gas wells aided by machine learning

- Developing a new rigorous drilling rate prediction model using a machine learning technique

- Geomechanical parameter estimation from mechanical specific energy using artificial intelligence

- Prediction of permeability from well logs using a new hybrid machine learning algorithm

- Hybrid machine learning algorithms to enhance lost-circulation prediction and management in the Marun oil field

- Optimization of controllable drilling parameters using a novel geomechanics-based workflow

- Shear wave velocity prediction from petrophysical logs using MLP-PSO algorithm

- A new approach to mechanical brittleness index modeling based on conventional well logs using hybrid algorithms

4. It is important to clarify how the separation ratio for training and test data was determined. Please explain the rationale behind choosing this specific ratio.

Response: We appreciate the excellent suggestion.Our selection of the 4:1 training and testing data partitioning ratio has been made after careful consideration, taking into account factors such as the scale of the dataset, the complexity of the research problem, and the requirements of model performance. Given the relative largeness of the dataset, we have ensured the retention of a sufficient amount of test data for robust model performance evaluation. Moreover, owing to the relatively straightforward nature of the research problem and the adaptability of the machine learning model employed, there is no imperative for an overly large training set to achieve effective model learning. We have supplemented our revised manuscript with an explanation regarding the choice of data partitioning ratio.

5. In the captions of tables and figures, it should be explicitly specified for which category of data they are intended.

Response: Thank you for the valuable suggestions from the reviewer. We have reviewed all the captions of the figures, and we have made the necessary revisions to address the concern you raised regarding unclear data categories. You can find the updated figure captions in the revised manuscript.

6. For further analysis, I recommend considering the following articles as valuable references:

Response: We sincerely appreciate your valuable recommendation.. We have thoroughly reviewed the literature and incorporated additional references related to machine learning analysis methods in the revised manuscript.

- Combined machine-learning and optimization models for predicting carbon dioxide trapping indexes in deep geological formations

- Predicting uniaxial compressive strength from drilling variables aided by hybrid machine learning

- Hybridized machine-learning for prompt prediction of rheology and filtration properties of water-based drilling fluids

- Machine-learning models to predict hydrogen uptake of porous carbon materials from influential variables

- Machine-learning predictions of solubility and residual trapping indexes of carbon dioxide from global geological storage sites

7. Among the various criteria considered, it is essential to mention in the text which specific criterion was used to select the best model.

Response: Thank you for the valuable feedback from the reviewer. In the revision, we will explicitly state that R2 will be highlighted as the primary performance evaluation metric for our model, emphasizing its crucial role in model performance assessment. At the same time, we will retain the use of RMSE, MAE, and MAPE, among other metrics, to provide a more comprehensive performance evaluation. The combined use of these metrics will aid readers in gaining a more comprehensive understanding of the model's performance characteristics.

8. It would be beneficial to provide additional explanations regarding the figures related to SHAP (Shapley Additive exPlanations). Please elaborate on the specific details and insights that can be derived from these figures.

Response: We thank the valuable suggestions. We have implemented a revision in section 3.2 - Feature Selection, where we included a more detailed explanation of the SHAP value analysis compared to the original version. This revised section delves into additional insights that can be derived from SHAP plots. You can find the exact locations of these changes in the revised manuscript.

Reviewer #2: 

Please find the review below:

1) Statistical analysis no present or weak lacks plots (histograms, heat map etc)

Response: We thank the valuable suggestions. In response, we have incorporated histograms for the statistical analysis of both resistivity and TCR in the data set in the third chapter. These histograms provide readers with a visual understanding of the distribution of resistivity and TCR values. We hope that this addition will offer readers a clearer insight into our data set.

2) Entire work is not novel. Why is ML used for this, simply adding ML to existing work has no new science or value

Response: Thank you for your comment, and we appreciate the concerns you've raised about the novelty of the research. We would like to provide further clarification. While machine learning is not a new technology in scientific research, our research aims to develop a new electrically conductive metal material that requires high levels of both electrical resistivity and TCR (Temperature Coefficient of Resistance). On the one hand, there is limited literature available on TCR studies for metal materials, especially for titanium alloys, as existing research has primarily focused on electrical resistivity. On the other hand, conducting experimental tests on a wide range of alloy materials is extremely challenging, making it practically unfeasible.

Therefore, we have chosen to employ machine learning predictive modeling based on available empirical data for the electrical performance of titanium alloys and the JMatPro database. This approach allows us to explore optimization strategies within the high-dimensional space of alloy elements, which is a crucial aspect of our research. We will emphasize this in the paper to ensure a clear understanding of our research objectives and significance.

3) Entire document needs to be rewritten, introduction lack's structure (research question, the knowledge gap and how this method is only one to fill that gap)

Response: Thank you for your valuable suggestion. We acknowledge that this is an important concern within the document, and we have revised the introduction to present the research question and knowledge gap more clearly, enabling readers to better comprehend the research background and objectives. In addition to the introduction, we have also made efforts to refine the other sections of the paper, avoiding lengthy paragraphs where possible.

4)TCR normalized per unit area plotted against length is needed

Response: We thank the valuable suggestions from the reviewer. In the beginning of Chapter 2, we introduced the method of calculating TCR. TCR's variation is solely dependent on the material itself and temperature. As a result, TCR variation curves only appear in the section where elemental adjustments are discussed. Due to the limited range of elemental adjustments, the TCR variation curves are also limited. We apologize for not being able to present TCR's temperature-dependent changes independently, as our entire study is based on predicting the TCR of titanium alloys at 25°C. However, to enhance the reader's experience, we have made some detailed improvements to the visualization of the elemental adjustment section, addressing previous issues related to overlapping curves.

5) Not sure why Al and Zr was choosen, did you measure composition changes of device using XPS/EDX for materials analysis, need proofs for that

Response: Thank you for your valuable comments. Our research primarily focuses on algorithm design and model development, aiming to address the prediction of material properties. The choice of these two elements resulted from calculation on the composition effects on the electrical resistivity and TCR in Titanium and the analysis of our feature importance algorithm. By employing a method that utilizes the intersection of features through two feature selection algorithms, we ultimately determined that aluminum and zirconium were the most significantly influential elements in the electrical performance of the titanium alloy studied in this research.

In this context, we did not employ experimental techniques such as XPS/EDX to measure changes in device composition considering the focus of this work is composition designing of new material. However, your commendation is very enlightening for us, and the verification of composition in the new alloy will be conducted in our near future work.

6) Very weak electro thermal tasting analysis requires more results (for ex R vs temperature plots, thermal conductivity measurements etc.)

Response: Thank you for your valuable comment. Our research primarily focuses on algorithmic modeling for new material design, and the original electrical performance data is sourced from the JMatPro database. Regarding your mention of thermal conductivity measurements, these may be conducted after we design new electrothermal metallic materials. As for the relationship graph between resistance and temperature that you mentioned, we have provided a set of data examples in the second section of Chapter 2 to illustrate the relationship between TCR and resistivity. We have since redesigned the image in question, aiming to provide readers with an improved reading experience.

7) Figures are not presented in proper way and looks like a master's thesis document or Lab manual

Response: Thank you for the reviewer's comments. We have made several adjustments to the majority of the figures and tables, including their presentation, background, fonts, colors, and positioning. In addition, we have incorporated new visualizations into the regression performance section of the predictive model.

---

## [Decision Letter · Decision Letter 1]

16 Jan 2024

Development of new materials for electrothermal metals using data driven and machine learning

PONE-D-23-26687R1

Dear Dr. He,

We’re pleased to inform you that your manuscript has been judged scientifically suitable for publication and will be formally accepted for publication once it meets all outstanding technical requirements.

Kind regards,

Babatunde Abiodun Salami

Academic Editor

PLOS ONE

Additional Editor Comments (optional):

Comments from PLOS Editorial Office: We note that one or more reviewers has recommended that you cite specific previously published works in previous decisions. As always, we recommend that you please review and evaluate the requested works to determine whether they are relevant and should be cited. It is not a requirement to cite these works, and you may remove any that you have added in response to the reviewer suggestions. We appreciate your attention to this request.

Reviewers' comments:

Reviewer's Responses to Questions

**Comments to the Author**

1. If the authors have adequately addressed your comments raised in a previous round of review and you feel that this manuscript is now acceptable for publication, you may indicate that here to bypass the “Comments to the Author” section, enter your conflict of interest statement in the “Confidential to Editor” section, and submit your "Accept" recommendation.

Reviewer #2: All comments have been addressed

2. Is the manuscript technically sound, and do the data support the conclusions?

Reviewer #2: Partly

3. Has the statistical analysis been performed appropriately and rigorously? 

Reviewer #2: Yes

4. Have the authors made all data underlying the findings in their manuscript fully available?

Reviewer #2: Yes

5. Is the manuscript presented in an intelligible fashion and written in standard English?

Reviewer #2: Yes

6. Review Comments to the Author

Reviewer #2: (No Response)

7. PLOS authors have the option to publish the peer review history of their article (what does this mean?). If published, this will include your full peer review and any attached files.

Reviewer #2: No
